# Spectrum of Causative Mutations in Patients with Hemophilia A in Russia

**DOI:** 10.3390/genes14020260

**Published:** 2023-01-19

**Authors:** Olesya Pshenichnikova, Valentina Salomashkina, Julia Poznyakova, Daria Selivanova, Daria Chernetskaya, Elena Yakovleva, Oksana Dimitrieva, Elena Likhacheva, Farida Perina, Nadezhda Zozulya, Vadim Surin

**Affiliations:** 1Laboratory of Genetic Engineering of National Medical Research Center for Hematology, Novy Zykovski Lane 4a, 125167 Moscow, Russia; 2Coagulopathies Department of National Medical Research Center for Hematology, Novy Zykovski Lane 4a, 125167 Moscow, Russia; 3Center for Pediatric Oncology and Hematology of State Autonomous Healthcare Institution ‘Sverdlovsk Regional Children’s Clinical Hospital’, Serafima Deryabina Street 32, 620149 Ekaterinburg, Russia

**Keywords:** hemophilia A, genetic diagnostics, pathogenic variant, *F8* gene

## Abstract

Hemophilia A (HA) is one of the most widespread, X-linked, inherited bleeding disorders, which results from defects in the *F8* gene. Nowadays, more than 3500 different pathogenic variants leading to HA have been described. Mutation analysis in HA is essential for accurate genetic counseling of patients and their relatives. We analyzed patients from 273 unrelated families with different forms of HA. The analysis consisted of testing for intron inversion (inv22 and inv1), and then sequencing all functionally important *F8* gene fragments. We identified 101 different pathogenic variants in 267 patients, among which 35 variants had never been previously reported in international databases. We found inv22 in 136 cases and inv1 in 12 patients. Large deletions (1–8 exons) were found in 5 patients, and we identified a large insertion in 1 patient. The remaining 113 patients carried point variants involving either single nucleotide or several consecutive nucleotides. We report herein the largest genetic analysis of HA patients issued in Russia.

## 1. Introduction

Hemophilia A (HA, MIM no. 306700) is an inherited, recessive, X-linked bleeding disorder caused by a wide spectrum of mutations in the gene encoding coagulation factor VIII (*F8* gene). HA affects 1 in 5000 males. The *F8* gene has a span of approximately 186 kb on chromosome X at locus q28 and consists of 26 exons [1]. The FVIII protein consists of a signal peptide (19 residues) and a sequence of domains (A1a1A2a2Ba3A3C1C2) that contains 2332 residues, for a total of 2351 residues. The mature FVIII molecule is a heterodimer, it circulates as heavy (A1A2B domains) and light (A3C1C2 domains) chains bound non-covalently by a divalent metal bridge. Cleavage, prior to secretion and at activation, results in a coagulant heterotrimer consisting of domains A1 + A2 + A3C1C2 [2]. Depending on the plasma procoagulant level of FVIII (FVIII:C, %), HA is classified into three clinical phenotypes: severe (FVIII:C <1%), moderate (FVIII:C 1–5%) and mild (FVIII:C >5%).

HA was the first inherited disease to be controlled by replacement treatment (i.e., the infusion of blood or blood products containing FVIII), but some patients develop antibodies against therapeutic FVIII, called inhibitors, which seriously compromise the patient’s prognosis. Understanding the factors that predispose a patient to such an adverse reaction is important for the management of HA [3,4,5,6].

More than 3500 different disease-causing pathogenic variants of HA have been identified and reported in international databases: Factor VIII Variant Database (f8-db.eahad.org (accessed on 25 December 2022)), Human Gene Mutation Database (www.hgmd.cf.ac.uk (accessed on 25 December 2022)), and CHAMP (https://www.cdc.gov/ncbddd/hemophilia/champs.html (accessed on 25 December 2022)). Among the gene defects that cause the severe form of hemophilia A, the most common defects include inversion of intron 22 (inv22) with a frequency of 40–50% and inversion of intron 1 (inv1) with a frequency of 2–5%. Inv22 is a result of intrachromosomal homologous recombination between the 9.5 kb region within intron 22 and one of two extragenic copies [7], and inv1 results from homologous recombination between inversely oriented sequences of 1.2 kb located in intron 1 and outside of the *F8* gene [8]. Other types of causative gene defects include missense and nonsense mutations, frameshift mutations, splice sites mutations, large insertions, and deletions. According to the databases, the majority of reported pathogenic variants are unique, i.e., reported only for one patient worldwide.

Mutation analysis in HA is essential for accurate genetic counseling of patients and their relatives. Unfortunately, this disease has a high degree of mutational heterogeneity that complicates carrier and prenatal diagnostics because it cannot be based on the screening of a limited number of common mutations (see f8-db.eahad.org (accessed on 25 December 2022)), except for the aforementioned inv22 and inv1. Nevertheless, there is a need for recording the population-wide spectrum in international mutation databases. This will allow comparative studies of the mutation frequencies in different populations, which may lead to identifying founder effect mutations and mutagenic hotspots characterized for separate populations. Mutation analysis of a given population is also useful for further understanding the correlation between structural and functional aspects of the mutant protein. The discovery rate of new gene defects leading to HA is growing exponentially [9,10,11,12,13,14,15,16,17,18,19,20,21,22,23,24,25,26].

However, in Russia, mutation analysis for HA is conducted only in a few laboratories. In our center, we have performed this analysis since 2015. To date, the only published work on the spectrum of *F8* mutations in Russian patients is *Complex Molecular Diagnostics of Haemophilia A in Russian Patients* [27], which included 117 patients. The aim of our study was to significantly increase the representativeness of the Russian HA population, describe the mutation spectrum of Russian patients with HA, and improve the genetic diagnostics for female carriers and prenatal testing of this disease in our country. All of the above is especially important for families with the sporadic form of HA (i.e., families with no known HA history) because only genetic analysis can give a reliable carrier status for the proband’s mother in these cases.

## 2. Materials and Methods

### 2.1. Patient Samples

In this study, we included patients from 273 unrelated families with different forms of HA, recruited from 1990 to 2022. They originated from different regions of Russia, and not all of them consulted with hematologists at our center. That is why clinical and laboratory data (including FVIII clotting activity and presence of inhibitors) were not available for all patients. The available patient information is provided in Appendix A.

As we frequently did not have an exact information about FVIII clotting activity, all severe and moderate patients were grouped together (severe/moderate patients).

We obtained material from the affected male proband in 252 families and from only the asymptomatic female carrier in 21 families.

The study was carried out according to the Principles of the Declaration of Helsinki and informed consent was obtained from all participants. Patients were considered to have HA according to the international consensus of the 2001 International Society on Thrombosis and Haemostasis (ISTH) Factor VIII and Factor IX Subcommittee [28].

### 2.2. DNA Collection and Extraction

Genomic DNA was isolated from EDTA-treated whole blood samples using phenol–chloroform extraction and ethanol precipitation [29]. Genomic DNA was dissolved in TE buffer and frozen until genotyping.

### 2.3. Detection of Mutations

#### 2.3.1. PCR Detection of Intron 1 and 22 Inversion

The severe/moderate patients were examined for inv22 and inv1. Inv22 was detected by modified long-range polymerase chain reaction (LD-PCR) [30] using Promega GoTaq^®^Long PCR Master Mix (Promega Corporation, Madison, WI, USA). Later in the study, we began using the method described in ref. [31], because it was more reliable. Inv1 was detected using an established method [8].

#### 2.3.2. Amplification of the F8 Gene

For patients without inversion of introns 1 and 22, we sequenced the entire *F8* coding region, including all exons and flanking intronic regions using primers developed in our laboratory [32]. The PCR reactions were carried out on a Tercik™ programmable thermocycler (DNK-Technology, Moscow, Russia) with PCR Master Mix (Thermo Fisher Scientific, Waltham, MA, USA) using 10 pmol of each oligonucleotide primer (Syntol, Moscow, Russia) and 50–100 ng template DNA in 25 μL reaction mixture. We analyzed the obtained PCR fragments with electrophoresis in 6% polyacrylamide gel (PAAG) or in 0.75% agarose gel (for LD-PCR products), followed by staining with ethidium bromide and visualization under UV light. Amplified DNA fragments were purified using the Wizard^®^ PCR Preps DNA Purification System (Promega Corporation, Madison, WI, USA) and subjected to direct cycle sequence analysis using the ABI PRISM^®^BigDye^TM^ Terminator v.3.1 Cycle Sequencing Kit (Thermo Fisher Scientific, Waltham, MA, USA) on an ABI PRISM 3100Avant genetic analyzer sequencer (Applied Biosystems, Foster City, CA, USA) at Genome CCU (Institute of Molecular Biology, Russian Academy of Sciences, Moscow, Russia).

#### 2.3.3. Large Deletions/Insertions Detection

Large *F8* gene deletions were identified by consistent failure of PCR amplification of a single exon or adjacent *F8* exons, as indicated by missing or altered bands upon electrophoresis of PCR products. At least three separate attempts to amplify missing fragments from the subject’s genomic DNA were performed using the same primers and PCR conditions, alongside successful amplification and sequencing of the exons flanking the suggested deletion. For detection of deletion breakpoints, we carried out LD-PCR using primers for amplification of the exons framing the deletion and GoTaq^®^ Long PCR Master Mix following the manufacturer’s protocol. We were able to identify deletion breakpoints for two patients. To accomplish this, we designed primers flanking the copies of the Alu element that could be involved in the deletion formation. For patient A375, we used primers F8-delF (5′-GTTTGTTTACATTTGTCCCAACT-3′, c.787+2045_2067, intron 6) and F8-delR (5′-TGCAACTCAAAGGACTAAACA-3′, c.1903 +1569_1589, intron 12). For patient A469, we used primers F8-5D [32] and Del6R (5′-CAGTTGACTCTTGAACAATACA-3′, c.787+2976_2995, intron 6).

Large duplications that could not be identified using routine PCR-based analysis methods were tested with multiplex ligation-dependent probe amplification (MLPA). MLPA was carried out using the *F8* SALSA MLPA kit P178 (MRC Holland, Amsterdam, The Netherlands) according to the manufacturer’s instructions. Exon dosage was calculated using Coffalyser.Net software (MRC Holland).

#### 2.3.4. Pathogenic Variant Evaluation

Missense and splice site mutations unreported in databases were examined using deleteriousness prediction scoring programs: PolyPhen-2 v2.2.2 [33], PROVEAN v.1.1.5 [34], SIFT v.6.2.1 [35], MutationTaster (build NCBI 37/Ensembl 69; [36]), CADD v1.6 [37], HApredictor [38], VarSite [39], and Missense3D [40]. Variants effects were then classified using the ACMG/AMP Variant Curation Guidelines [41].

#### 2.3.5. Variant Nomenclature

The cDNA numbering system was compliant with the Human Genome Variation Society recommendations ver. 15.11 (http://varnomen.hgvs.org (accessed on 25 December 2022)). Amino acid numbering was based upon the start methionine, as codon +1. The reference sequence was NG_011403.2 for genomic positioning and NM_000132.4 for cDNA numbering. As reference databases for pathogenic variants, we used the Factor VIII Variant Database (f8-db.eahad.org (accessed on 25 December 2022)), Human Gene Mutation Database (www.hgmd.cf.ac.uk (accessed on 25 December 2022)), and CHAMP (https://www.cdc.gov/ncbddd/hemophilia/champs.html (accessed on 25 December 2022)).

## 3. Results

Molecular analysis of the *F8* gene in 273 unrelated patients allowed us to identify pathogenic variants in 267 patients (97.8%). In the remaining six patients (2.2%), we did not find causal variants in the *F8* gene. One patient had pathogenic variants in the *vWF* gene, so his diagnosis was changed from HA to von Willebrand disease (vWD) type 2N [42]. The remaining five patients did not have pathogenic variants in the *vWF* gene.

Among the 267 patients with successfully identified *F8* gene alterations, we found 101 different pathogenic variants, 35 of which had never been previously reported in international databases. A summary of the identified *F8* gene defects and clinical features of patients are presented in Appendix A. The distribution of different pathogenic variant types in our sample population is given in Figure 1. We found inv22 in 136 cases (50.9% of patients with genetically verified HA) and inv1 in 12 patients (4.5%). Large deletions spanning one to eight exons were found in five patients (1.9%), and we identified a large insertion in one patient (0.4%). The remaining 113 (42.3%) patients carried point variants involving either single nucleotides or several consecutive nucleotides.

### 3.1. Inversions

We found inv22 in 136 out of 267 unrelated patients (50.9%, Table 1), the majority of whom had severe/moderate HA (122 out of 124, 98% of cases with known HA severity) and 20 of whom had developed inhibitors.

We found inv1 in 12 out of 267 patients (4.5%, Table 1). Patients with inv1 also had predominantly severe/moderate HA (11 out of 12, 91.7% of cases with known HA severity) and one patient had developed inhibitors. Notably, three patients had abnormal inv1 with additional deletions or duplications of adjacent regions; these patients were described in more detail in ref. [43] along with several similar cases from our population.

### 3.2. Large Deletions and Insertions

In five patients with severe/moderate HA, we found large deletions. They were initially identified by failure to amplify certain exons and then confirmed using MLPA.

All five large deletions were unique and affected different exons (Appendix A): ex2–6 with breakpoints in introns 1 and 6; whole ex6 with breakpoints in introns 5 and 6 (Figure 2a); ex7–12 with breakpoints in introns 6 and 12 (Figure 2b); ex15–22 with breakpoints in introns 14 and 22; and ex23–25 with breakpoints in introns 22 and 25. In two cases, we were able to identify the exact breakpoints. In the remaining three cases, the introns involved in the deletion formation (introns 1, 14 and especially intron 22) were extremely long and GC-enriched, making the breakpoint identification technically challenging.

In one patient, PCR of exon 14 of the *F8* gene yielded a fragment approximately 1500 bp longer than expected from the reference sequence. Sanger sequencing allowed us to identify a large insertion between nucleotides c.3117 and c.3118.

For the exon 6 deletion, we implemented LD-PCR using primers flanking exons 5 and 7. As a result, we obtained a fragment of 15,000 bp instead of the normal fragment of 18,000 bp. We suggested that the deletion might be caused by homologous recombination between different copies of Alu elements. Our calculations showed that the obtained length of the PCR product could be achieved if the involved Alu element from intron 6 was AluYe5 (c.787 + 2221 − c.787 + 2484). Therefore, we specifically designed primer Del6R flanking this Alu element to use it as a reverse primer in the PCR system with the forward primer flanking exon 5. PCR yielded a fragment nearly 2000 bp in length, while the distance between the primers in the whole *F8* gene was approximately 5600 bp. Sequencing of this fragment enabled us to determine the deletion breakpoints. One deletion breakpoint was located in intron 5 (c.671 − 1051) between two Alu elements: AluJb (c.670 + 656 − c.670 + 948) and AluSz (c.671 − 659 − c.671 − 351). Another deletion breakpoint (c.787 + 2580) was close to abovementioned AluYe5 element in intron 6, inside AluSg element (c.787 + 2496 − c.787 + 2781) in intron 6 (Figure 2a,c). Using the same PCR system with primers F8-5D/Del6R, we also detected this deletion in family members of the patient. His mother and sister appeared to be carriers of this gene defect, while his cousin was not.

In the case of the ex7–12 deletion, we used LD-PCR with the exon 6 forward primer and exon 13 reverse primer (distance between the primers in the whole *F8* gene was approximately 37 000 bp) and obtained the only PCR fragment that was nearly 7500 bp in length. As in the previous case, we suggested that the deletion might involve Alu elements. We chose the same AluYe5 in intron 6 (c.787 + 2221 − c.787 + 2484) and AluY (c.1903 + 1051 − c.1903 + 1348) in intron 12 since our calculations showed that a fragment 7500 bp in length could be obtained in LD-PCR only if those copies of Alu-repeat were involved in the formation of the deletion. We specifically designed primers F8-delF and F8-delR flanking those Alu elements. Amplification of the patient’s DNA using those primers yielded a fragment approximately 1300 bp in length. Sequencing of this fragment enabled us to determine the deletion breakpoints. One of the deletion breakpoints was 273 bp upstream from the AluYe5 element in intron 6 (c.787 + 1949) and another deletion breakpoints was inside the AluY element in intron 12 (position c.1903 + 1113) (Figure 2b,c). Using the PCR system with primers F8-delF/F8-delR, we also detected this deletion in family members of the patient, where seven out of nine women appeared to be carriers of this gene defect.

### 3.3. Loss-of-Function Variants

We revealed 35 different variants leading to the loss of protein function in 48 out of 267 patients (17.9% of the sample). Among them there were 20 single nucleotide substitutions, 6 microdeletions (1–2 nucleotides), 4 microduplications (1–2 nucleotides), and 5 indels.

Two out of 5 indels resulted from the simple substitution of two consecutive nucleotides, while 3 indels were more complex in nature. Among the three observed complex indels, one case was associated with triplication of a 12-nucleotide fragment, but deleted and inserted sequences did not have anything in common in the remaining two cases (Figure 3).

Microduplications and 3 indels of a complex nature resulted in frameshifts, summing up in 13 different variants in 22 out of 267 patients (8.2% of the sample population).

Two indels with two consecutive nucleotide changes and 20 single nucleotide substitutions resulted in nonsense mutations. In total, they affected 26 out of 267 patients and represented 22 different pathogenic variants (9.7% of the sample population).

Five nonsense and eight frameshift variants were not previously described.

The majority of patients with loss-of-function mutations had severe/moderate HA (39 out of 42 patients, 92.8% of cases with known severity). Two patients with frameshift mutations and 7 patients with nonsense mutations had developed inhibitors (Table 1).

### 3.4. Splicing Variants

In 10 out of 267 unrelated patients (3.7%), we revealed 10 different genetic alterations affecting splicing. Eight out of ten variations were changed canonical splicing dinucleotides (±1–2 positions from an exon), one substitution was in the +4 position, and the last variation affected the +5 position. Three splicing variants were not previously reported. In another paper exploring Russian HA patients, the authors described splicing mutations c.6901-1G>C and c.6901-2A>C [27], which were also identified in our patients but have not been described in other countries.

All 9 patients with splicing mutations, for whom we knew F8:C (%), had severe/moderate HA and one patient had developed inhibitors (Table 1).

### 3.5. Missense Mutations and Inframe Deletion

We identified 44 different missense variants in 56 out of 267 patients (20.2%), 16 of which were not previously described.

Among patients with missense variants, those with severe/moderate HA (29 out of 50 patients, 58% of cases with known severity) were slightly more prevalent. However, this was obviously a result of our sample population being skewed towards severe/moderate HA and not the characteristics of the Russian population. Two patients with missense mutations (one with severe/moderate HA and another with mild HA) had developed inhibitors (Table 1).

In our sample population, one inframe deletion (0.4%) was detected—known pathogenic variant c.5142_5144delACG p.(Arg1715del) [44]. Although the influence of this type of genetic alterations on protein function is usually disputable, it was evaluated by all deleteriousness prediction scoring methods as pathogenic. Variant p.(Arg1715del) was identified in a patient with severe HA without data about the presence of inhibitor antibodies.

### 3.6. Previously Undescribed Variants Assessment

Among the studied patients, we identified 29 variants that affected one to several nucleotides and had not been previously reported (Table 2). This included 4 deletions of 1–2 nucleotides, a nucleotide duplication, 19 substitutions of a single nucleotide, and five indels. These variants resulted in frameshift (N = 8), nonsense (N = 5), splicing (N = 2), and missense (N = 14) mutations. Only one of the new variants was recurrent.

According to the ACMG/AMP Variant Curation Guidelines [41], we classified 24 out of 29 variants as pathogenic and likely pathogenic, while a verdict of uncertain significance was obtained for 5 missense variants. Deleteriousness prediction software results are given in Appendix A.

All large deletions and the large insertion were also previously undescribed, but their pathogenicity was undoubted.

## 4. Discussion

### 4.1. Clinical Manifestations of Studied Patients

We had clinical data for the majority of our patients; however, we were unable to determine the severity of HA for 23 out of 267 families (8.6%) and no information about the presence of inhibitors was available for 165 cases (61.3%). Most families had the severe/moderate form of HA—217 cases (81.3% of all 267 HA patients, 88.9% of all 244 patients with known severity). This made our sample population strongly skewed towards severe/moderate HA, which is sometimes observed in studies [9,44] despite the proportions of severe/moderate/mild HA in the population believed to be approximately 50%/10%/40% [45]. Indeed, in broader studies, especially those based on national registers, these expected proportions are often met [16,46,47,48]. The prevalence of more clinically prominent cases in our data can be explained by the specifics of patient recruitment. Patients with severe HA experience more everyday inconveniences than those with milder forms of HA, and therefore are more interested in genetic diagnostics, including female relative carrier detection.

The prevalence of severe forms of HA in our sample population was concordant with the observed proportions of inv1 and inv22 that jointly covered approximately 55% of the sample [7,8].

### 4.2. Patients without Genetic Variants in F8 Gene

We were unable to find the genetic cause of the clinical presentation in five cases out of 273 (1.8%). For those patients, we excluded common inversions (inv1 and inv22), nucleotide substitutions in all *F8* gene exons and adjacent intronic regions, large deletions, and insertions. Our methods did not include mRNA analysis and whole gene sequencing, so we cannot exclude deep intronic variants leading to alternative splicing, pathogenic variants in regulatory regions, or unique inversions that could not be detected by Sanger sequencing and MLPA.

HA patients without detectable *F8* gene defects or with pathogenic variants in *vWF* instead of *F8* are common. Such patients may comprise 1–10% of the sample population when only DNA analysis is available [9,20,21,22,44,46,47,48,49,50], whereas mRNA analysis enables the identification of causative variants in more patients [16]. Usually, analysis of mRNA leads to the identification of deep intronic mutations (point substitutions or large intronic deletions) or synonymous substitutions leading to the formation of alternative splicing sites [16,51,52,53,54,55,56,57]; however, in rare cases, mRNA analysis also provides no results [16,53,58]. This leaves the possible explanations to be regulatory mutations or mutations in other genes that either cause an HA-like phenotype (e.g., *LMAN1* or *MCFD2*) or encode proteins interacting with FVIII [58]. The possible role of *F8* mRNA 3′UTR-targeting miRNAs was shown in a FVIII deficiency phenotype clinically manifesting as HA [59].

It also worth noting that although deep intronic mutations are sometimes identified in patients with severe or moderate HA [16,52,57], they are generally found in patients with the milder form of HA [51,53,55,57]. In contrast, in this study, four out of five patients without identified causative variants had severe HA (FVIII:C < 1%).

### 4.3. Patients with Two Genetic Variants

Two unrelated patients (A429, A384) simultaneously had two variants in the *F8* gene. The DNA of probands’ mothers was available, so we verified that these variants did not appear de novo.

In patient A429, we identified two variants that have never been previously described: a single nucleotide substitution leading to the missense amino acid change p.(Pro1265Leu) and a substitution of two consecutive nucleotides (classified as indel according to the HGVS guidelines) leading to the nonsense amino acid change p.(Met1363Ile*). The missense variant was located 98 codons before the premature termination codon. However, the p.(Met1363Ile*) variant clearly had a predominant influence on the patient’s phenotype, so it is impossible to say whether the p.(Pro1265Leu) variant was pathogenic or benign. Moreover, this missense variant was evaluated by most of the used deleteriousness prediction scoring methods as benign.

In patient A384, we identified two substitutions both leading to missense amino acid changes: p.(Arg301Cys) reported in the EAHAD database as pathogenic and p.(His336Arg), which was new. Both missense variants were predicted to be pathogenic according to the PolyPhen-2, MutationTaster, SIFT, and CADD scoring programs, but VarSite and Missense3D yielded different results. P.(Arg301Cys) was evaluated as likely damaging: the change in amino acid sidechain size being large, Arg>Cys substitution—very highly unfavored, changing the buried charged amino acid with an uncharged amino acid, and the p.301 position being highly conserved. P.(His336Arg) was more tolerated: the change in amino acid sidechain size not being large; His>Arg substitution—neutral, without any structural consequences, and the p.336 position being fairly conserved. According to ACMG/AMP guidelines, p.(His336Arg) was classified as having an uncertain significance (Table 1). To sum up, all predictions led us to the suggestion that the patient’s phenotype was influenced by p.(Arg301Cys), while p.(His336Arg) seemed likely to be benign.

Therefore, despite the fact that we identified two patients with more than one genetic alteration in the *F8* gene, we cannot confirm with certainty that both variants in both cases were pathogenic.

### 4.4. Localization of Point Variants

As was reported in other studies [9,10,11,12,13,14,15,16,17,18,19,20,21,22,23,24,25,26], single nucleotide variations (SNVs) were evenly distributed over the *F8* gene exons with the exclusion of missense mutations being virtually absent at the fragment corresponding to the B domain (Figure 4).

Among our patients with SNVs, we identified 12 recurrent variants found in 2–7 patients apiece and 78 unique variants found in a single patient each. Deletions, duplications, and substitutions were represented among recurrent as well as among unique variants. All indels were unique. Deletions and duplications were slightly more represented among recurrent variants than among unique variants—25% (3 out of 12) and 17% (13 out of 78), respectively.

Among the unique variants were 65 single nucleotide substitutions (18 new), 5 deletions (4 new), three duplications (1 new), and 5 indels (all new).

The recurrent variants were 9 single nucleotide substitutions (1 new), two known deletions, and one known duplication. We compared haplotypes of patients with recurring variants using polymorphisms in the *F8* gene and showed that only in one substitution occurred a founder effect which we were able to prove [32], while the remaining cases were independent appearances.

Seven out of nine recurring nucleotide substitutions were CpG substitutions, all of which were previously described. Another two recurring substitutions (1 new) were not located at CpG sites, both of which were found only in the Russian population. Fourteen out of 65 unique substitutions were in CpG sites, all of which were previously described. The remaining 41 unique substitutions (16 new) were not in CpG sites.

It is widely believed that CpG sites are hotspots for mutagenesis and recurrent mutations usually occur in these locations, but in fact, not all CpG are equally mutagenic, which is also true for the *F8* gene (Appendix A). In our sample population, only 21 out of 74 SNV (28.4%) were in CpG sites. CpG sites involved in mutagenesis differ in their activity according to the Factor VIII Variant Database. Some CpG sites are extremely active (e.g., p.(Arg612Cys)—patients; p.(Ala723Thr)—144 patients; p.(Arg2016Trp)—100 patients, Appendix A), while other sites are almost inactive (e.g., p.(Arg245Trp)—1 patient; p.(Glu739Lys)—8 patients; p.(Arg1715Gln)—3 patients, Appendix A), and some sites have intermediate activity (e.g., p.(Glu291Lys)—patients; p.(Arg355Stop)—44 patients; p.(Arg1715Stop)—23 patients, Appendix A). Notably, all substitutions in CpG sites identified in our population fell mostly into the intermediate activity group but were not the most active CpG sites (according to the Factor VIII Variant Database; Appendix A). These results suggest that our HA population has a unique pattern of CpG mutagenesis.

### 4.5. Large Deletions

There are three main types of molecular mechanisms of genomic rearrangements: replication-based mechanisms (RBMs), non-allelic homologous recombination (NAHR), and non-homologous non-replicative DNA repair mechanisms. RBMs result from replication slippage and template switching during DNA replication and produce mostly microhomology at breakpoint junctions. NAHR between two genomic regions with high sequence homology (>99%) results in extensive sequence homology at breakpoint junctions. Examples of NAHR are the most prevalent mutations in HA—inv22 and inv1. There are two types of non-homologous non-replicative DNA repair mechanisms: non-homologous end joining (NHEJ) results mostly in blunt ends with sometimes a short insertion of random nucleotides at breakpoint junctions, and alternative end-joining (Alt-EJ), which can generate short microhomologies. Large deletions in the *F8* gene usually appear as a result of NHEJ [60].

In our case, both large deletions with identified breakpoints resulted from NHEJ involving Alu repeats, but in both cases, only one of the breakpoints was exactly in an Alu repeat, while another breakpoint was outside it (Figure 2c). Interestingly, in both cases, one of the breakpoints was located in intron 6, which comprised two Alu repeats—AluSg and AluYe5. This genomic region appeared to be a DNA breakage hotspot that was previously noted by other researchers [60].

### 4.6. De Novo Origin Evaluation

The probability of de novo origin of the identified variants in probands was evaluated on the basis of family HA history and direct verification of the identified variants in female relatives.

In the case of a family history of HA, de novo origin of the genetic variants in our patients was excluded. The presence of family HA history was verified if family members were able to identify another HA patient in the same or earlier generations. In all other cases, family history was classified as “no data” because knowledge about relatives beyond the third generation in our country is frequently absent due to the prevalence of nuclear families.

Another way to clarify the status of mutation was to test for the presence of the mutation in female carriers of the same or ascending generations (e.g., mother or sister of proband). If the mutation was identified in these relatives, then de novo origin was excluded. If the mutation in the same or ascending generations was absent, then de novo origin was confirmed. If only descending female relatives (e.g., daughter of proband) or no female carriers were available, then the mutation origin in family was deemed undeterminable.

We were able to assess de novo origin of the pathogenic variants in 159 out of 267 apparently unrelated patients. In 81 families, de novo origin was excluded due to family history, and it was ruled out in 73 families because of verified mutation carrier status in female relatives. We confirmed de novo origin in only 5 cases, as patients’ mothers did not carry *F8* gene variants identified in the proband. This resulted in the proportion of de novo pathogenic variants in HA to be 3.15% (5 out of 159). Five confirmed de novo origin cases involved four single nucleotide variants (two nonsense and two missense amino acid changes) and one complex indel.

### 4.7. Inhibitor Development

The numbers of patients with different pathogenic variant types, including their clinical presentation and inhibitor history, are summarized in Table 2. We had data about inhibitor status for 102 patients, 37 of whom had developed inhibitors (36.3%). Thirty six patients with inhibitors had severe/moderate HA. This prevalence was concordant with the literature data [16,61,62].

Integration of data obtained by different research groups led to the following distribution of the types of mutations in the *F8* gene in relation to the risk of inhibitor development:High risk of inhibitor development: large deletions (several exons), nonsense mutations in the light chain (A3C1C2 domains);Moderate risk: large deletions (one exon), nonsense mutations in the heavy chain (A1A2B domains), inv22, inv1;Low risk: microdeletions/microinsertions, missense mutations, splicing mutations [63].

Overall, our results corresponded with published data [3,4,5,63]. Patients with large deletions were in the high-risk group (4 out of 5, or 80% had developed inhibitors). According to the classification, nonsense mutations are present in the high- and moderate-risk groups, which held true for our sample population. Additionally, as in the literature data, it seems that nonsense mutations in the heavy chain of FVIII protein were rarely associated with the development of inhibitors (5 out of 7, or 71% of patients without inhibitors had nonsense mutations in this chain), compared to nonsense mutations in the light chain. Patients with inv22 and inv1 were in the intermediate-risk group. The frequency of inhibitor development in this group did not differ significantly from the overall frequency in the HA population. Missense mutations occurred in the low-risk group, as only 7.7% (2 out of 26) of patients with this mutation type had developed inhibitors. As for the influence of frameshift and splicing mutations on inhibitor development, this could not be evaluated owing to the lack of inhibitor status information for those groups.

## 5. Conclusions

We report herein the largest genetic analysis of HA patients issued in Russia. Although the overall mutation spectrum of the *F8* gene in the Russian population reflected the tendencies revealed in other populations, some interesting characteristics were also found, such as a different activity pattern of CpG sites, a mutation with a founder effect, splicing mutations that have been already described twice in the Russian population but nowhere else, etc. Nevertheless, this data could improve the genetic diagnostics for female carriers and prenatal testing of this disease in our country. Russia has a vast and complex geographic distribution and history, so, undoubtfully, this study will need to be continued in order to increase the coverage of HA patients.

## Figures and Tables

**Figure 1 genes-14-00260-f001:**
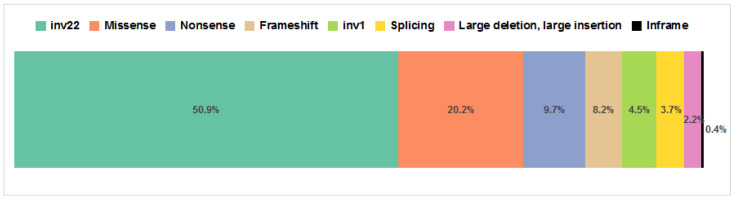
Proportions of different pathogenic variant types found among 267 patients from Russia.

**Figure 2 genes-14-00260-f002:**
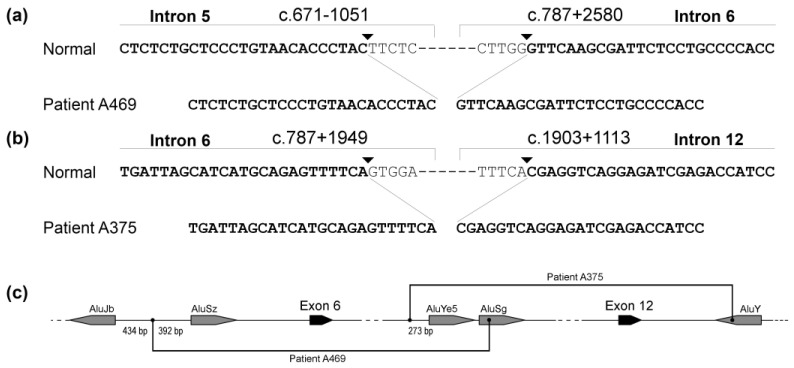
Breakpoints for two large deletions: (**a**) Deletion of exon 6; (**b**) Deletion of exons 7–12; (**c**) Locations of established breakpoints relative to adjacent Alu elements.

**Figure 3 genes-14-00260-f003:**
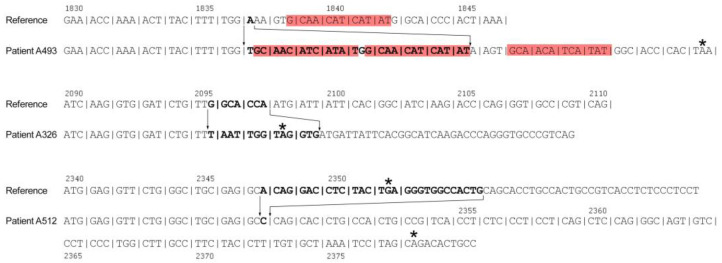
Nucleotide sequences of three new complex indels in the *F8* gene. Red boxes indicate triplicated nucleotides, * indicates stop codon.

**Figure 4 genes-14-00260-f004:**
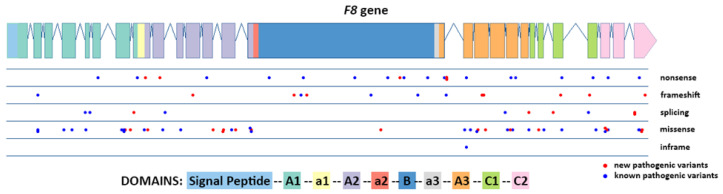
Distribution of different pathogenic variant types found among 267 patients from Russia, except for inversions and large deletions/insertion.

**Table 1 genes-14-00260-t001:** Summary of severity and inhibitor data in patients from this study.

	Severe/Moderate				Mild				ND ^1^		Total
	Inh	No Inh	ND	Total	Inh	No Inh	ND	Total	ND	Total	
Inv22	20	23	79	122			2	2	12	12	136
Inv1	1	6	4	11			1	1		0	12
Large deletions	4	1		5				0		0	5
Large insertions		1		1				0		0	1
Frameshift	2	2	12	16			3	3	3	3	22
Nonsense	7	7	9	23				0	3	3	26
Splicing	1	1	7	9				0	1	1	10
Missense	1	15	13	29	1	9	11	21	4	4	54
Inframe			1	1				0		0	1

^1^ ND—no data.

**Table 2 genes-14-00260-t002:** Previously undescribed variants in the *F8* gene. SNV—single nucleotide variant; M—missense; S—splicing; N—nonsense; F—frameshift.

Pathogenic Variant	Nucleotide Change Type	AA Effect	N Patients	Pathogenicity [41]
c.926C>T p.(Pro309Leu)	SNV	M	1	Uncertain significance (PM2, PP2, PP3)
c.1007A>G p.(His336Arg)	SNV	M	1	Uncertain significance (PM2, PP2, PP3, PP4, BP2)
c.1010-2A>G	SNV	S	1	Pathogenic (PVS1, PM2, PP3, PP4)
c.1195A>T p.(Lys399*)	SNV	N	1	Pathogenic (PVS1, PM2, PP4)
c.1366A>T p.(Lys456*)	SNV	N	1	Pathogenic (PVS1, PM2, PP4)
c.1633delC p.(Pro545Leufs*4)	Deletion	F	1	Likely pathogenic (PVS1, PM2)
c.1911T>A p.(Asn637Lys)	SNV	M	1	Likely pathogenic (PM1, PM2, PM5, PP2, PP3, PP4)
c.1921T>C p.(Phe641Leu)	SNV	M	1	Likely pathogenic (PM2, PM5, PP2, PP3)
c.2156G>T p.(Arg719Ile)	SNV	M	1	Pathogenic (PS2, PM2, PM5, PP2, PP3, PP4)
c.2830delA p.(Lys943Serfs*12)	Deletion	F	1	Pathogenic (PVS1, PM2, PP4)
c.3031_3032del p.(Lys1011Serfs*1)	Deletion	F	1	Likely pathogenic (PVS1, PM2)
c.3794C>T p.(Pro1265Leu)	SNV	M	1	Uncertain significance (PM2, PP2, PP4, BP2, BP4)
c.4089_4090delinsTT p.(Met1363Ile*)	Indel	N	1	Pathogenic (PVS1, PM2, PP4)
c.4834A>T p.(Lys1612*)	SNV	N	1	Likely pathogenic (PVS1, PM2)
c.4836_4837delinsAT p.(Lys1613*)	Indel	N	1	Pathogenic (PVS1, PM2, PP4)
c.5413T>C p.(Tyr1805His)	SNV	M	1	Likely pathogenic (PM2, PM5, PP2, PP3)
c.5475dupT p.(Val1826Cysfs*4)	Insertion	F	1	Likely pathogenic (PVS1, PM2)
c.5509delinsTGCAACATCATATGGCAACATCATAT p.(Lys1837Cysfs*17)	Complex indel	F	1	Pathogenic (PVS1, PS2, PM2)
c.5533A>G p.(Thr1845Ala)	SNV	M	2	Uncertain significance (PM2, PM5, PP2, BP4)
c.5878C>G p.(Arg1960Gly)	SNV	M	1	Likely pathogenic (PM1, PM2, PM5, PP2, PP3)
c.6193T>A p.(Trp2065Arg)	SNV	M	1	Pathogenic (PS1, PM2, PM5, PP2, PP3)
c.6274-2A>G	SNV	S	1	Pathogenic (PVS1, PM2, PP3, PP4)
c.6285_6291delinsTAATTGGTAGGTG p.(Leu2095Phefs*3)	Complex indel	F	1	Pathogenic (PVS1, PM2, PP4)
c.6442delA p.(Asn2148Metfs*7)	Deletion	F	1	Pathogenic (PVS1, PM2, PP4)
c.6634T>G p.(Ser2212Ala)	SNV	M	1	Likely pathogenic (PM2, PM5, PP2, PP3)
c.6664T>C p.(Trp2222Arg)	SNV	M	1	Likely pathogenic (PS1, PM2, PP2, PP3, PP4)
c.7013T>C p.(Leu2338Pro)	SNV	M	1	Uncertain significance (PM2, PP2, PP3, PP4)
c.7021G>C p.(Glu2341Gln)	SNV	M	1	Likely pathogenic (PM2, PM5, PP2, PP3, PP4)
c.7041_7068delinsC p.(Asp2349Hisfs*29)	Complex indel	F	1	Likely pathogenic (PVS1, PM2)

## Data Availability

Data is available on request from the authors.

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
