# Peer review of "Spectrum of Causative Mutations in Patients with Hemophilia A in Russia"

_genes, 2023, doi:10.3390/genes14020260_

Round 1

Reviewer 1 Report

The research investigated pathogenic variants of patients from 273 unrelated families. It’s a valuable resource that has been collected for over 30 years. This nice research performed some fundamental works that will contribute to HA genetic counselling field. The research question in this manuscript is well addressed with concrete experiment and analysis, also in a clear written style. I only have a few suggestions and questions as following:

1.     The manuscript often uses percentage to describe ratio of cases under certain scenario, however, it is sometimes hard to understand where the value comes from. For example, in line 171, “98% of cases with known HA severity”, I cannot figure out how this number is calculated from table1. Can the author also provide the exact number for numerator and denominator?

2.     Can you provide the gender for the 273 samples. If all investigated proband with severe HA are male?

3.     What is the normal population control used in this study? So, a disease causative variant can be distinguished from a benign one generated from normal polymorphism?

Author Response

Point 1. The manuscript often uses percentage to describe ratio of cases under certain scenario, however, it is sometimes hard to understand where the value comes from. For example, in line 171, “98% of cases with known HA severity”, I cannot figure out how this number is calculated from table1. Can the author also provide the exact number for numerator and denominator?

Response 1. Thank you for the remark, we added exact numbers where they were not reported throughout the text.

Point 2. Can you provide the gender for the 273 samples. If all investigated proband with severe HA are male?

Response 2. Thank you for the question. Among them 252 samples were from affected males and 21 samples were from asymptomatic female carriers. This information was on the lines 86-87, but we added there some details.

Point 3. What is the normal population control used in this study? So, a disease causative variant can be distinguished from a benign one generated from normal polymorphism?

Response 3. Thank you for the remark. Unfortunatelly, we did not use control population. Due to the facts, that F8 gene has a span of about 186 kb and there is almost none known SNP in coding regions, we would require a great cohort as a control. That is financially and technically extremely difficult.

For all patients without inversions we sequenced all functionally important regions of F8 gene. When clinical HA phenotype was present and no other  variants except the described one were found, we could conclude that found variant was causative. 

Reviewer 2 Report

The authors conducted an extensive analysis of F8 mutations in Russian HA patients. I have a few questions:

- If globally over 3500 pathogenic variants have been described, and in this study, 101 different pathogenic variants have been found in 267 Russian patients, do you expect many more undetected variants to be found in the future when more patients are sequenced in Russia?

- (section 3.2): Alu elements are frequent throughout the genome, and are suggested to function as sites for homologous recombination (HR) in the F8 gene. Is Alu-mediated HR known (or has it been suggested) to occur in other genes that lead to other genetic diseases? If not, is there evidence for increased presence of Alu elements in F8 compared to their presence in other genes?

- would you suggest/recommend whole-gene sequencing for each patient?

- Minor: line 326-327 'did not appeared' should be: had not appeared (or: did not appear)

Author Response

Point 1. If globally over 3500 pathogenic variants have been described, and in this study, 101 different pathogenic variants have been found in 267 Russian patients, do you expect many more undetected variants to be found in the future when more patients are sequenced in Russia?

Response 1. Thank you for the question. This is quite possible, because one of HA characteristics is that there are a lot of unique mutations. As for mild HA, we can also find more variants with a founder effect in isolated regions of our country.

Point 2. (section 3.2): Alu elements are frequent throughout the genome, and are suggested to function as sites for homologous recombination (HR) in the F8 gene. Is Alu-mediated HR known (or has it been suggested) to occur in other genes that lead to other genetic diseases? If not, is there evidence for increased presence of Alu elements in F8 compared to their presence in other genes?

Response 2. Thank you for the question. Actually, Alu-mediated HR leading to inherited diseases is quite usual event, it was described in Pulmonary arterial hypertension (BMPR2 gene), Peeling skin disease (CDSN gene), Fanconi anemia (FA gene), Gaucher disease (GBA1 gene) etc. (see also https://doi.org/10.5808/GI.2016.14.3.70).

Point 3. would you suggest/recommend whole-gene sequencing for each patient?

Response 3. Thank you for the question. It seems to be unjustified, as only a small percent of patients has pathogenic variants outside exons or exon/intron junctions (using common algorithm we did not find causing variants in 6 out of 273 patients), so we would reccomend whole-gene sequencing only when all other options was exhausted.

Point 4. Minor: line 326-327 'did not appeared' should be: had not appeared (or: did not appear)

Response 4. Thank you a lot, we have editted it.

Reviewer 3 Report

This paper reports mutations of the coagulation factor VIII (F8) gene identified in patients with Hemophilia A (HA) disease. Among 273 patients from unrelated Russian families, 267 were found to contain 101 pathogenic F8 variants by analyzing genomic DNA sequences using PCR and Sanger sequencing methods. Strikingly, 35 variants were not previously reported but were highly likely HA causative. In addition, the authors of this paper discussed the failure of F8 variants identification in 5 patients due to possible limitations of the applied and DNA-based methods. More importantly, the authors discussed mRNA-based sequencing methods as potential alternatives for the diagnosis of F8 variants in these patients. The strength of this paper is that it analyzed the largest HA genetic cohort in Russia for genetic variations linked to F8. This study provides HA diagnostic and parental testing references not only in Russia but also worldwide.

1.     What’s the frequency of mutation co-occurrence in one patient/family? For Figure 1, a pie chart also indicates overlapped variants will be helpful to present the data as it also shows the frequency of mutation co-occurrence if any.

2.     The resolution of Figure 2 is low, it’s better to improve it.

3.     Do any nonsense mutations identified in this cohort related to known transcription/splicing/translation defects, if not, is there any discussion on the possible mechanisms linking these mutations to HA?

4.     (Optional) Description of the number of mutations in 3.1-3.6 are not very straightforward for first-time readers due to high mutation diversity. For each mutation group in 3.1-3.6, a simplified table at the beginning of each section would be helpful.

Author Response

Point 1. What’s the frequency of mutation co-occurrence in one patient/family? For Figure 1, a pie chart also indicates overlapped variants will be helpful to present the data as it also shows the frequency of mutation co-occurrence if any.

Response 1. Thank you for the remark, we hope that we correctly understand the question. HA is a X-linked recessive disease and almost all patients had only one mutation, except for two people, which are discussed in details in section 4.3. Figure 1 could be a pie chart, but we suppose that a linear model is similarly informative, so we would like not to change it. Due to the fact that almost all patients had only one mutation (as was mentioned above) in the process of analysis and visualization we considered each patient only once. As for two patients with two genetic variants in F8 we took into account only one of them which was supposed as more pathogenic (in one patient it was missense mutation, in another - nonsense).  

Point 2. The resolution of Figure 2 is low, it’s better to improve it.

Response 2. Thank you, we improved it.

Point 3. Do any nonsense mutations identified in this cohort related to known transcription/splicing/translation defects, if not, is there any discussion on the possible mechanisms linking these mutations to HA?

Response 3. Thank you for the question.  All five new nonsense mutations were located in the beginning or the middle of the gene and led to nonfunctional significantly truncated protein or to the removing of the protein by the mechanism of the nonsense-mediated decay. There is a possibility that nonsense mutations located in the most distal part of the gene would not affect gene's functionality. However in our case they were all known and described for the world population. 

Point 4. (Optional) Description of the number of mutations in 3.1-3.6 are not very straightforward for first-time readers due to high mutation diversity. For each mutation group in 3.1-3.6, a simplified table at the beginning of each section would be helpful.

Response 4. Thank you for the idea, but we are afraid, that it would overload the text, and would dublicate information in the article and in tables, all information about found variants is given in Supplementary.